# Breastfeeding after Returning to Work: A Systematic Review and Meta-Analysis

**DOI:** 10.3390/ijerph18168631

**Published:** 2021-08-15

**Authors:** Frédéric Dutheil, Grégory Méchin, Philippe Vorilhon, Amanda C. Benson, Anne Bottet, Maëlys Clinchamps, Chloé Barasinski, Valentin Navel

**Affiliations:** 1CNRS, LaPSCo, Physiological and Psychosocial Stress, University Hospital of Clermont-Ferrand, CHU Clermont-Ferrand, Occupational and Environmental Medicine, Université Clermont Auvergne, WittyFit, F-63000 Clermont-Ferrand, France; maelysclinchamps@gmail.com; 2Department of General Practice, UFR Medicine, 28 Place Henri-Dunant, Université Clermont Auvergne, F-63000 Clermont-Ferrand, France; gregory.mechin@gmail.com; 3Department of General Practice, UFR Medicine, Research Unit ACCePPT Self-Medication, Multi-Professional Support for Patients, Université Clermont Auvergne, 28 Place Henri-Dunant, F-63000 Clermont-Ferrand, France; pvorilhon2@wanadoo.fr (P.V.); anne.bottet@uca.fr (A.B.); 4Swinburne University of Technology, Health and Biostatistics, Hawthorn, Victoria, VIC 3122, Australia; abenson@swin.edu.au; 5CNRS, SIGMA Clermont, Institut Pascal, University Hospital of Clermont-Ferrand, CHU Clermont-Ferrand, Université Clermont Auvergne Perinatality, F-63000 Clermont-Ferrand, France; cbarasinski@chu-clermontferrand.fr; 6CNRS, INSERM, GReD, CHU Clermont-Ferrand, University Hospital of Clermont-Ferrand, Ophthalmology, Université Clermont Auvergne, F-63000 Clermont-Ferrand, France; valentin.navel@hotmail.fr

**Keywords:** lactation, occupation, public health, pregnancy, well-being

## Abstract

Background: The benefits of breastfeeding are widely known; however, continuation after returning to work (RTW) is not. We aimed to conduct a systematic review and meta-analysis to assess the prevalence of breastfeeding after RTW. The secondary objectives were to compare the economic statuses between continents. Method: PubMed, Cochrane Library, Base, and Embase were searched until 1 September 2020, and two independent reviewers selected the studies and collated the data. To be included, articles needed to describe our primary outcome, i.e., prevalence of breastfeeding after RTW. Results: We included 14 studies, analyzing 42,820 women. The overall prevalence of breastfeeding after RTW was 25% (95% CI, 21% to 29%), with an important heterogeneity (I^2^ = 98.6%)—prevalence ranging from 2% to 61%. Stratification by continents and by GDP per capita also showed huge heterogeneity. The Middle East had the weakest total prevalence with 10% (6% to 14%), and Oceania the strongest with 35% (21% to 50%). Despite the prevalence of breastfeeding in general increasing with GDP per capita (<US$5000: 19%, US$5000–30,000: 22%; US$30,000 to 50,000: 25%, >US$50,000 42%), the prevalence of non-exclusive breastfeeding follows more of a U-curve with the lowest and highest GDP per capita having the highest percentages of breastfeeding (<US$5000: 47% and >US$50,000: 50%, versus <28% for all other categories). Conclusion: Breastfeeding after RTW is widely heterogeneous across the world. Despite economic status playing a role in breastfeeding after RTW, cultural aspects seem influential. The lack of data regarding breastfeeding after RTW in most countries demonstrates the strong need of data to inform effective preventive strategies.

## 1. Introduction

Breastfeeding provides multiple health advantages for the child (infections, malocclusion, and intelligence) and their mother (breast cancer) [1,2,3,4], with economic and social benefits as well (cost savings for parents, employers, and society, as well as the parent–child relationship) [3,5,6,7]. Hence, the World Health Organization (WHO) recommends “exclusive breastfeeding for the first 6 months of life and introduction of nutritionally-adequate and safe complementary (solid) foods at 6 months together with continued breastfeeding up to 2 years of age or beyond” [8]. During this breastfeeding transition time, returning to work (RTW) is common for mothers who have to manage work and breastfeeding. RTW represents one of the main reasons for stopping breastfeeding [9,10,11,12]. Combining breastfeeding and work may be hard for mothers depending on their working conditions [13], sociocultural heritage and gender role ideology [14], public health policies [15], and economy and lobby groups [16]. For example, in a Taiwanese study, 67% of working mothers initiated breastfeeding, but only 10% continued after RTW [17]. Both the culture of work and breastfeeding differ between countries; for example, breastfeeding initiation may vary from 47% (Ireland) to 99% (Norway) [18] within developed European countries. In addition to breastfeeding initiation, the type of breastfeeding (exclusive or non-exclusive) may also be at the interplay between the work environment and sociocultural/economic aspects [19]. However, no studies have summarized the differences in breastfeeding after RTW or have compared countries. Conversely, women from low-income countries have difficulty combining work and breastfeeding [20], and therefore might be at risk of ceasing breastfeeding when returning to work. Considering the importance of breastfeeding, an evidence-based study is needed to summarize the existing literature for building efficient promotion and support for breastfeeding in the workplace. 

Therefore, we aimed to conduct a systematic review and meta-analysis to evaluate the prevalence of breastfeeding after RTW (primary aim). The secondary objectives were to evaluate the differences between continents or their level of development, as well as putative influencing variables such as sociodemographics [21,22,23], breastfeeding support at work [24,25,26,27], or workplace policy [28,29,30]. Additionally, we evaluated the influence of the previous factors on the type of breastfeeding (exclusive or not).

## 2. Methods

### 2.1. Literature Search

We reviewed all studies involving breastfeeding after returning to work. Specifically, the inclusion criteria for the search strategy were the prevalence of breastfeeding and/or exclusive breastfeeding after RTW, using the following keywords: Breastfeeding AND work (see detailed search strategy in Section A.1). The following databases were searched on 1 September 2020: PubMed, Cochrane Library, Embase. and Base. The search was not limited to specific years. To be included, articles needed to describe our primary outcome variable, which was the prevalence of breastfeeding after RTW, i.e., women had to have returned to work and studies had to have reported the timing of RTW. Specifically, we excluded studies when mothers did not work, or did not describe breastfeeding and its timing related to RTW. Studies that were not written in English or French were also excluded, as well as qualitative studies. In addition, reference lists of all publications meeting the inclusion criteria were manually searched to identify any further studies not found through electronic searching. The PRISMA flow diagram of the search strategy is presented in Figure 1. Two authors (G. Méchin and M. Clinchamps) conducted all of the literature searches, as well as collated and independently reviewed the abstracts. Based on the selection criteria, they decided the suitability of the articles for inclusion. A third author (F. Dutheil) was asked to review the articles where consensus on suitability was debated. Then, all authors reviewed the eligible articles. We followed the PRISMA guidelines (Section A.2) [31].

### 2.2. Data Collection

The data collected included the authors’ name, publication year, study design, duration of studies, aims, outcomes of the included articles, sample size, mean age, occupation, countries and continents, and their economic status (gross domestic product (GDP) per capita), month of RTW, breastfeeding practices (global, exclusive, or non-exclusive), and characteristics of the individuals (such as education, birth delivery, and smoking) (Table 1).

### 2.3. Quality of Assessment

An assessment of the methodological quality was performed using the Newcastle–Ottawa Scale (NOS) for cohort studies [32] and modified NOS for cross-sectional studies [33]. The items assessed were selection bias (four items), comparability bias (one item), and outcome bias (three items for cohort and two for cross-sectional studies). Each item was assigned a judgment of “Yes” (1 point), “No” (0 point), or “Can’t say” (0 point). Thus, the maximum score was 8 points for cohort studies and 7 points for cross-sectional studies (Section A.3 and Section A.4). Disagreements between reviewers (G. Méchin and M. Clinchamps) were addressed by obtaining a consensus with a third author (F. Dutheil).

### 2.4. Statistical Considerations

Statistical analysis was conducted using Stata software (version 15, StataCorp, College Station, TX, USA) [34,35,36,37,38,39,40,41]. The characteristics of breastfeeding, work, the individuals, or other variables were summarized for each study sample and reported as the mean ± standard deviation (SD) and number (%) for continuous and categorical variables, respectively. Random effects meta-analyses (DerSimonian and Laird approach) on the prevalence of breastfeeding after RTW were conducted when the data could be pooled [42]. *p*-Values less than 0.05 were considered statistically significant. We stratified these meta-analyses by continents and by economic status of the countries (GDP per capita). All of these meta-analyses were computed for global, exclusive, and non-exclusive breastfeeding. Heterogeneity in the study results was evaluated by examining forest plots and confidence intervals (CIs) and by using formal tests for homogeneity based on the I^2^ statistic, which is the most common metric for measuring the magnitude of between-study heterogeneity and is easily interpretable. I^2^ values range between 0% and 100% and are typically considered low for <25%, modest for 25%–50%, and high for >50% [42]. For example, significant heterogeneity may be due to the variability between the characteristics of the studies, such as the type of breastfeeding (exclusive or not), occupational settings, or the characteristics of the countries or individuals. For thoroughness, funnel plots of these meta-analyses were used to search for potential publication biases. In order to verify the strength of the results, further meta-analyses were then conducted, excluding studies that were not evenly distributed around the base of the funnel [43]. When possible (sufficient sample size), meta-regressions were proposed to study the relationship between the prevalence of breastfeeding after RTW and putative variables such as continent, economic status of countries (GDP), or characteristics of the individuals (age, education, etc.). The results are expressed as regression coefficients and 95% CIs.

## 3. Results

An initial search produced a possible 1832 articles (Figure 1). Removal of duplicates (*n* = 383) and applying the selection criteria reduced these articles reporting the prevalence of breastfeeding after RTW to 14 studies (Figure 1) [44,45,46,47,48,49,50,51,52,53,54,55,56,57]. All of the identified articles were written in English (Table 1).

### 3.1. Quality of the Articles

The quality assessment of the 14 included studies, as outlined by the NOS, varied from 57.1% [57] to 100% [44], with a mean score of 81.8 ± 7.9%. The most frequent biases were the assessment of outcomes (self-reported) for cohort studies and the selection, especially considering the limited sample size in some studies. There was also a lack of follow-up in the cohort studies. Detailed characteristics of methodological quality assessment of each included study are available in Section A.3 and Section A.4. All studies mentioned ethical approval. 

### 3.2. Population

**Sample size**: Population sizes ranged from 84 [45] to 20,172 [49]. In total, 42,820 women were included in this meta-analysis.

**Age:** All studies reported age. Seven studies reported mean age [46,49,50,51,52,54,56], ranging from 26.9 [54] to 33 years [56], and seven studies reported a cut-off for age [44,45,47,48,53,55,57] from <25 to >30 years old. 

**Gender:** All studies included only women (42,820 in total).

**Type of occupation:** Nine studies included all working mothers with no job specification [46,47,48,49,50,52,53,54,55]. Two studies included employed workers in formal and informal sectors [51,57]. One study included mothers who were professional/semi-professional, manual, or business workers [44]. One study included mothers in paid employment [56]. One study included government and semi-government employees, private company employees, and self-employed or family business owners [45].

**Country of breastfeeding:** Two studies were conducted in Europe (France [47] and the United Kingdom [51]), two studies in the Middle East (Israel [46] and Egypt [44]), three in the United States of America [52,53,54], four in Asia (Thailand [45,57], India [48], and Taiwan [49]), and three in Oceania (Australia [50,55,56]).

**Gross domestic product per capita:** We retrieved the GDP per capita by country and year of the included studies using data from the World Bank database [58].

**Other characteristics:** Characteristics such as education [44,47,48,49,50,52,53,54,56,57], mode of delivery [44,45,48,49,50,52,53,55], and smoking status [47,50,52,53,55] were inconsistently reported, precluding further analyses (Table 1).

### 3.3. Inclusion and Exclusion Criteria within the Included Articles

Working mothers were the shared inclusion criterion for the 14 studies [44,45,46,47,48,49,50,51,52,53,54,55,56,57]. Six studies included working mothers who had regular work over the 12 months prior to birth [44,48,52,53,56,57]. Two studies specified that charitable work was excluded [45,49]. Two studies only included single mothers [51,53], with one restricting inclusion to British/Irish white natural mothers [51]. Two studies only included infants free of any serious health conditions [50,55]. Three studies only included mothers who initiated breastfeeding [47,52,54] prior to RTW. The exclusion criteria were a severe illness, either in the mother or the baby [45,48,49], mothers of twins [44], and mothers who never initiated breastfeeding [56].

### 3.4. Outcome and Aim of the Studies

The primary outcome of the included articles was the prevalence of breastfeeding after RTW for six studies [44,46,49,50,53,56], and the duration of breastfeeding for two studies [47,51]. The other studies aimed to assess the factors related to breastfeeding at work [45,48,52,54,55,57].

### 3.5. Study Designs

Seven studies had a cross-sectional prevalence survey design, analyzing breastfeeding amongst working mothers [44,45,46,48,50,56,57]. Seven studies had a cohort follow-up design [47,49,51,52,53,54,55], analyzing the prevalence of breastfeeding after RTW over time [47,49,51,53,55] or from survey data [52,54].

### 3.6. Breastfeeding and Return to Work

**Method of assessment:** Breastfeeding after RTW was measured via a questionnaire [47,50], semi-structured interview questions [44,45,48,49,57], telephone [55,56], or at home [46,51,53]. Two studies retrieved breastfeeding prevalence using survey data at follow-up [52,54].

**Type of breastfeeding:** Seven studies investigated both exclusive and non-exclusive breastfeeding [47,50,53,54,55,56,57], only two reported exclusive breastfeeding [44,48], and only five reported non-exclusive breastfeeding [45,46,49,51,52].

**Return to work:** RTW after birth ranged from <1 month [49] to 12 months [49,50,55]. The heterogeneous time of RTW precluded stratification of breastfeeding by month of RTW (Table 1).

### 3.7. Meta-Analysis on the Prevalence of Breastfeeding after Returning to Work

Our meta-analysis demonstrated an overall prevalence of breastfeeding after RTW of 25% (95% CI, 21% to 29%), with an important heterogeneity (I^2^ = 98.6%)—the prevalence of breastfeeding after RTW ranging from 2% [48] to 61% [45]. Stratification by continents (Section A.5) and by GDP per capita (Section A.6) also showed large heterogeneity. Middle Eastern countries had the weakest total prevalence with 10% (6% to 14%), and Oceania (Australia) the strongest with 35% (21% to 50%). The prevalence of breastfeeding was 19% (10% to 28%) for GDP under US$5000 per capita, 22% (18% to 26%) between US$5000 and US$30,000, 25% (18% to 32%) between US$30,000 and US$50,000, and 42% (24% to 60%) for GDP higher than US$50,000 (Figure 2).

Similarly, the meta-analysis on exclusive and non-exclusive breastfeeding showed high heterogeneity (I^2^ > 90%), with a mean overall prevalence of breastfeeding after RTW of 21% (14% to 28%) and 28% (23% to 32%), respectively. Stratification by continents demonstrated similar results, with Middle Eastern countries having the weakest prevalence (5% (3% to 7%) and 14% (8% to 19%), respectively) and Oceania countries the strongest (26% (4% to 47%) and 42% (29% to 55%), respectively). Stratification by GDP did not show an increase in exclusive or non-exclusive breastfeeding with the economic status of countries. For example, for non-exclusive breastfeeding, the highest prevalence of breastfeeding was for the lowest and highest GDP (47% (41% to 54%) for GDP under US$5000 per capita and 50% (45% to 55%) for GDP higher than US$50,000, whereas the prevalence was 20% (17% to 22%) between US$5000 and US$30,000 and 28% (20% to 36%) between US$30,000 and US$50,000) (Figure 2). 

### 3.8. Sensitivity Analysis and Other Meta-Regressions

Funnel plots of these meta-analyses demonstrated a wide heterogeneity (Figure 3), precluding any sensitivity analyses, with most studies being outside of the meta-funnels. For overall and non-exclusive breastfeeding, meta-regressions by continent demonstrated a lower prevalence of breastfeeding in the Middle East compared to Asia (coefficient = 0.15, 95% CI = 0.02 to 0.29 and 0.19, 0.05 to 0.33, respectively) and Oceania (0.28, 0.13 to 0.42 and 0.23, 0.10 to 0.37, respectively) and was also higher in Oceania vs. Europe (0.18, 0.03 to 0.33 and 0.22, 0.07 to 0.37, respectively). The prevalence of overall breastfeeding was also lower in the Middle East compared to the United States of America (0.15, 0.03 to 0.28), and the prevalence of non-exclusive breastfeeding was also higher in Oceania vs. the United States of America (0.18, 0.05 to 0.30). The meta-regressions did not show any exclusive significant association by continent. For overall and non-exclusive breastfeeding, the meta-regressions demonstrated a higher prevalence of breastfeeding for the countries with the highest GDP (>US$50,000) than those with GDP between US$5000 and US$30,000 (0.20, 0.07 to 0.32 and 0.31, 0.2 to 0.41, respectively), between US$30,000 and US$50,000 (0.17, 0.03 to 0.3 and 0.22, 0.11 to 0.33, respectively), and <US$5000 (0.22, 0.07 to 0.37, but only for overall breastfeeding). For non-exclusive breastfeeding, those countries with the lowest GDP (<US$5000) also had a higher prevalence of breastfeeding than countries with GDP between US$5000 and US$30,000 (0.31, 0.15 to 0.48) and between US$30,000 and US$50,000 (0.23, 0.06 to 0.40). The meta-regressions did not demonstrate any influence of individual characteristics (age, education, etc.) (Figure 4).

## 4. Discussion

The main finding was that the prevalence of breastfeeding after RTW is widely heterogeneous across the world. Despite the review demonstrating that economic status may play a role in breastfeeding after RTW, cultural aspects seem an important determinant. We did not find an effect of putative influencing variables.

### 4.1. Breastfeeding around the World

This study is the first meta-analysis analyzing breastfeeding prevalence after RTW. As stated by the WHO, breastfeeding confers various benefits for infants and mothers [59]. However, RTW is one of the major causes (20%) of women stopping breastfeeding, along with fatigue (22%) and insufficient milk supply (21%) [60]. The intention to breastfeed is negatively associated with RTW [61]. The dominant trends of our meta-analysis were heterogeneity and lack of data. We demonstrated a huge heterogeneity in breastfeeding after RTW between and within continents. Even within industrialized European countries, comparisons between countries were available mainly for breastfeeding initiation and duration with a large heterogeneity. For example, France and the U.K. are among the countries with the lowest initiation (62% and 70%, respectively [62]) and prevalence at 12 months [3], whereas Scandinavian countries have the highest initiation (99% for Denmark and Norway [62]) and long-term prevalence. The results from our meta-analysis seemed to show a higher rate of breastfeeding after RTW in Asia than in Europe, in line with the literature (almost 100% of breastfeeding initiation in Myanmar, for example [63]). The United States of America seems to have a similar breastfeeding rate after RTW to Asia. Oceania, represented by Australia, has high rates of breastfeeding after RTW, in line with their goal by 2022 of 40% exclusive breastfeeding until newborns are six months old [64]. Middle Eastern countries have the lowest prevalence of breastfeeding after RTW, in line with their low breastfeeding initiation rate of only one-third of newborns, falling to 20% at six months, without considering returning to work [65]. Even if breastfeeding in general has been widely studied, we demonstrated that data are scarce regarding breastfeeding after RTW in most countries, particularly in some continents such as Africa, where no data are available, demonstrating the urgent need for data from these countries to inform effective preventive strategies. It is known that the cultural aspect is very important for breastfeeding uptake [19]. Mothers’ mothers have a strong positive attitude toward breastfeeding when they are positively reinforced or supported [66]. Notably, highly educated Chinese grandmothers were associated with decreased exclusive breastfeeding in their daughters [67]. This fact could be linked with gender role ideology that varies markedly across countries [68]. Moreover, social and cultural attitudes have an impact on the representation of breastfeeding within and between different countries/continents. A meta-analysis found that community-based interventions, including group counselling or education and social mobilization, with or without mass media, are effective at increasing timely breastfeeding initiation by 86% and exclusive breastfeeding by 20% [19].

### 4.2. Cultural Aspect in Breastfeeding

Interestingly, whatever their economic status, some countries have strong breastfeeding policies, especially after RTW [69]. Australia developed breastfeeding reference groups [70], maternity leave policies [70], and support clinics [71] with home visiting programs [72]. Maternity leave also positively impacts breastfeeding duration [10,12,73,74]. A recent review showed a positive relationship between maternity leave length and breastfeeding duration [75]. Australia, along with Austria and New Zealand, also have high female part-time employment [3], more compatible with breastfeeding after RTW [76]. Moreover, a recent study in Australia highlighted that women’s emotional well-being is related to breastfeeding [77], which may in turn improve well-being at work. In comparison, some developing countries are also culturally prone to breastfeeding, such as Thailand or Myanmar, who regularly promote breastfeeding support assistance after RTW [78,79]. Similarly, 50% of women continue to breastfeed until their child reaches two years of age in Laos and Indonesia, and almost 65% in Myanmar [63]. Our meta-analysis also suggested that exclusive breastfeeding is lower after RTW than non-exclusive breastfeeding. Not surprisingly, combining breastfeeding and work necessitates adaptation—such as the introduction of infant formula, which is very common in countries such as Indonesia [15]. The frequency of infant formula use in Asia may also explain the U-shape of the prevalence curve of non-exclusive breastfeeding (lowest and highest GDP per capita having the highest percentages of breastfeeding). Some working conditions, such as shift work, add difficulties for mothers to exclusively breastfeed their infant [13]. Breastfeeding can also be at the interplay between public health policies, the economy, and lobby groups. In the USA, the Infant Formula Council historically lobbied against the public health promotion of breastfeeding [16], even discouraging a pro-breastfeeding campaign in 2007 [80]. In 2009, only 23/50 states in the USA encouraged workplace breastfeeding by adopting laws, and no state required employers to provide breastfeeding pumping equipment to their employees [81]. In 2011, the USA ranked last out of 36 countries for its breastfeeding policy [16]. Eager to improve workplace lactation, the USA launched ambitious programs [82] that included reasonable break times and adequate space for nursing mothers to express milk [83]. More broadly, the Lancet breastfeeding series highlighted that the promotion of breastfeeding is a collective societal responsibility and not the sole responsibility of an individual woman [19]. One of the six call to action points was to foster positive societal attitudes toward breastfeeding, such as adequate maternity leave and the opportunity to breastfeed or express milk in the workplace [19].

### 4.3. Other Factors Influencing Breastfeeding after Returning to Work

We did not find other factors that influenced breastfeeding prevalence after RTW. Obviously, the literature showed that early RTW negatively affected breastfeeding initiation [84] and duration [73,85], as well as full-time work [10,30]. On the contrary, part-time work has been found to have a positive impact on breastfeeding duration [86,87]. Flexibility in working schedules may be associated with breastfeeding [88]. Despite no studies, the acceptance of teleworking following the COVID-19 pandemic could also help women to breastfeed [89]. Interestingly, the guarantee of paid breastfeeding breaks for at least six months has been shown to be associated with an increase of nearly 9% in exclusive breastfeeding [90]. Some workplace variables seem to be strongly associated with breastfeeding after RTW. Based on the literature, workplace support seems to be an important influence on breastfeeding duration after returning to work. Managerial and organizational support increases exclusive breastfeeding duration nearly twofold [91], with co-workers’ support being essential in the decision to continue breastfeeding [92]. Lack of breastfeeding facilities, such as a room dedicated for breastfeeding or a fridge, is associated with breastfeeding discontinuation after RTW [93]. Even if some laws promote breastfeeding at work, such as the Federal Break Time for Nursing Mothers law requiring employers covered by the Fair Labor Standards Act (FLSA) to provide basic accommodations for breastfeeding mothers at work in the USA, these laws are still not fully applied [16] and need to be expanded worldwide. There is very limited or even inexistent literature on the putative sociodemographic and clinical factors linked to breastfeeding after RTW. However, the literature is vast on factors known to affect breastfeeding initiation and duration. Mothers over 35 years old have higher chance of breastfeeding initiation [21] and continuation at six months [94]. Single parents or mothers without support from their partner have levels of lower initiation [95]. Smoking mothers are also less likely to initiate breastfeeding [96,97], as well as those who had a cesarean section [98] or those with lower education levels [22]. Cesarean section and low income are also two factors that decrease the duration of breastfeeding [98,99]. A recent study in Oceania also demonstrated that most of the previous factors are also risk factors for stopping full breastfeeding [100]. No data were found to indicate if infant sex influences breastfeeding initiation or duration. Multiparous women are more likely to breastfeed for six months or more [23], and by consequence are more likely to continue breastfeeding. Although controversial [101], among the other risk factors of non-breastfeeding are, research suggests, maternal obesity [94], not attending childbirth education [94], depression [94], or dyad connection [97].

## 5. Limitations

All meta-analyses have limitations [102]. Meta-analyses inherit the limitations of the individual studies of which they are composed and are subjected to a bias of selection of included studies. However, the use of broader keywords in the search strategy limited the number of missing studies. Despite our rigorous criteria for including studies in our meta-analysis, their quality varied. Most cross-sectional studies included in our meta-analyses described a bias of self-report, such as skipping questions and incomplete information, nondisclosure, and uncertainty regarding the timing of the questionnaire. Though there were similarities between the inclusion criteria, they were not identical. In particular, some studies included only mothers who worked the year before delivery, whereas other studies did not specify [47,49,50,54,55]. Two studies only included women who initiated breastfeeding [47,54], which may have led to a comparison bias; however, sensitivity analyses without these two studies did not influence the results. Moreover, our meta-analysis was based on a moderate number of studies, especially for exclusive breastfeeding. An important finding of our study is also the lack of breastfeeding data after RTW—some continents had no data available. Stratification by ethnicity was not feasible because of the lack of data; however, stratification by country/continent enabled international comparisons and should have taken into account the influence of baseline breastfeeding rates. Furthermore, the dates of RTW were too heterogeneous to stratify for; the lack of included studies also precluded stratification by time—both of which may have impacted the comparisons between continents and GDP.

## 6. Conclusions

Despite the scarcity of data, the prevalence of breastfeeding after returning to work is 25% and widely heterogeneous across the world. Even if economic status plays a role in breastfeeding after return to work, cultural aspects seem an important determinant, influencing public health policies and workplace breastfeeding support. We also showed the lack of data regarding breastfeeding after returning to work in most countries, with no data available from some continents such as Africa, demonstrating the strong need for data in these countries to inform effective preventive strategies.

## Figures and Tables

**Figure 1 ijerph-18-08631-f001:**
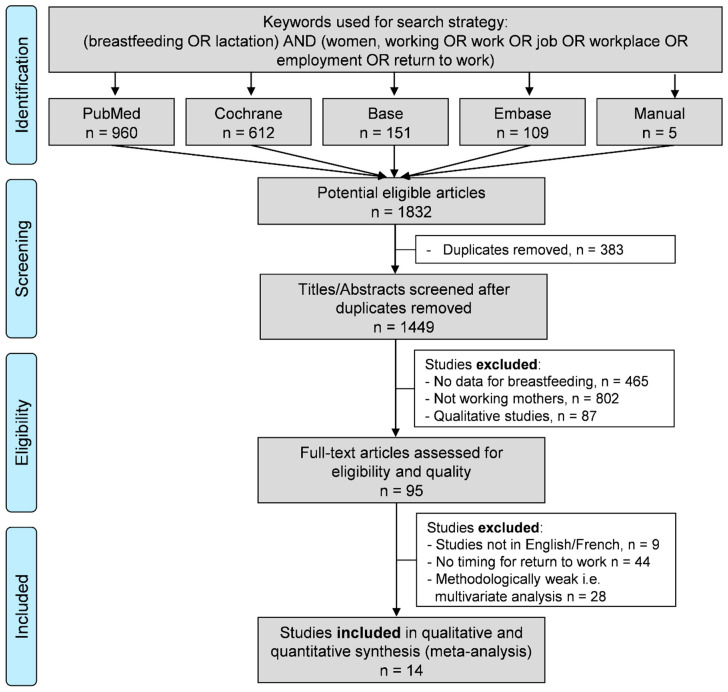
Search strategy.

**Figure 2 ijerph-18-08631-f002:**
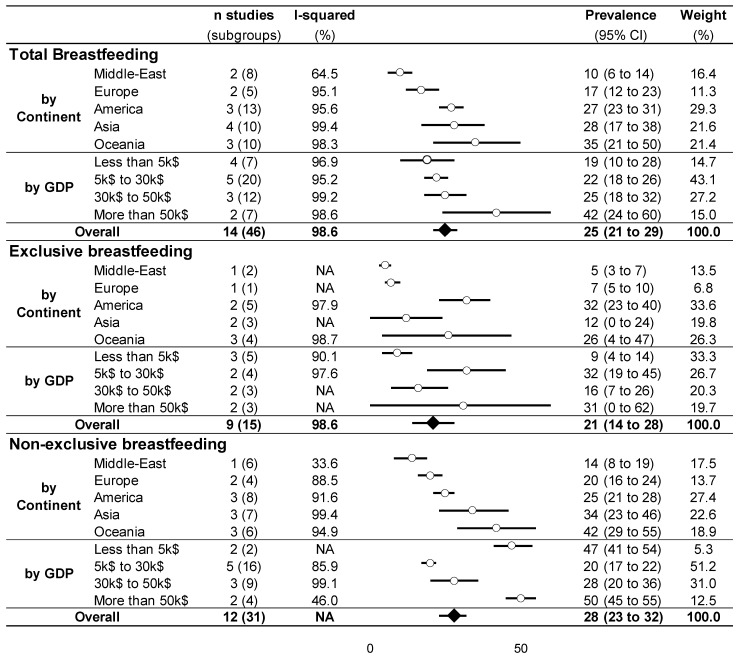
Meta-analysis of the prevalence of breastfeeding.

**Figure 3 ijerph-18-08631-f003:**
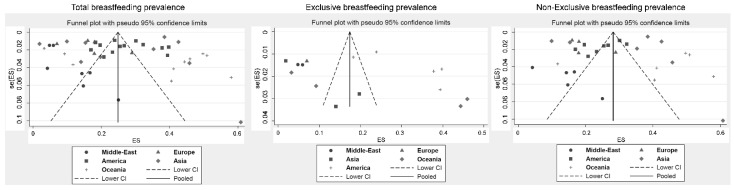
Meta-funnels.

**Figure 4 ijerph-18-08631-f004:**
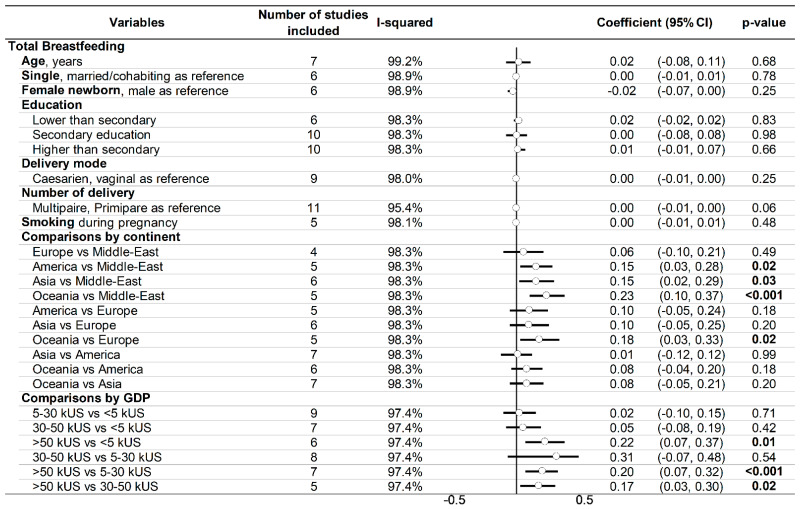
Meta-regression.

**Table 1 ijerph-18-08631-t001:** Characteristics of included studies. * Adjusted by years of the study.

Study	Country	Type of Study	Follow-Up	Population	Recruitment Procedures	Occupation	GDP per Capita * (in $)	Type of Breastfeeding	Timing of Returning to Work	Other Parameters Measured
**Abou-ElWafa 2019** [44]	Egypt	Cross-sectional study	July–December 2017	633	All working mothers attending healthcare facilities	Professional/semi-professional; manual; business worker	2413	Exclusive	<4 months; 4 months	Maternal sociodemographics, employment patterns, and birth characteristics
**Aikawa 2015** [45]	Thailand	Cross-sectional study	February 2008	84	Mothers who visited the breastfeeding mobile clinic at a nursery goods exhibition in Bangkok	Government and semi-government; private company employee; self-employed or family business owner	4379	Non-exclusive	<3 months	Maternal sociodemographics, employment patterns, and birth characteristics
**Bergman 1981** [46]	Israel	Cross-sectional study	1979	291	Working women interviewed 7–9 months after delivery	All workers	5674	Non-exclusive	<3 months; 3 months; 4 months; 4–5 months; 5 months; 6 months	Maternal sociodemographics and employment patterns
**Bonet 2013** [47]	France	Cohort study	2003–2006	979	From EDEN mother–child cohort; pregnant women were recruited from the maternity wards of the Poitiers and Nancy University hospitals	All workers	34,760	Exclusive and non-exclusive	≤4 months; 5–8 months	Maternal sociodemographics and employment patterns
**Boralingiah 2016** [48]	India	Cross-sectional study	January–December 2014	107	Working mothers of the children attending the immunization center at JSS Hospital, Mysuru	All workers	1576	Exclusive	<6 months; >6 months	Maternal sociodemographics, employment patterns, and hospital breastfeeding practice
**Chuang 2010** [49]	Taiwan	Cohort study	2006–2007	20,172	From the Taiwan Birth Cohort Study	All workers	30,100	Non-exclusive	≤1 month; ≤2 months; ≤3 months; ≤6 months; ≤12 months	Maternal sociodemographics, employment patterns, birth characteristics, and hospital feeding practices
**Cox 2015** [50]	Australia	Cross-sectional study	2010–2011	427	Mothers recruited from maternity services in rural western Australia	All workers	51,937	Exclusive and non-exclusive	<6 months; 6–12 months	Maternal sociodemographics, employment patterns, birth characteristics, hospital feeding practices, and psychosocial factors
**Hawkins 2007** [51]	U.K.	Cohort study	September 2000–January 2002	6917	From the Millennium Cohort Study	Employed workers in the formal or informal sector	27,427	Non-exclusive	<3 months; 4 months	Maternal sociodemographics and employment patterns
**Jacknowitz 2008** [52]	USA	Cohort study	1989–1999	1506	From the National Longitudinal Survey of Youth and the Children of the National Longitudinal Survey	All workers	24,405	Non-exclusive	<6 weeks; >6 weeks and ≤3 months; >3 months and ≤6 months	Maternal sociodemographics, employment patterns, and birth characteristics
**Ogbuanu 2011** [53]	USA	Cohort study	2001–2003	6150	Data drawn from the Early Childhood Longitudinal Study–Birth Cohort	All workers	39,677	Exclusive and non-exclusive	<6 weeks; <3 months	Maternal sociodemographics, employment patterns, birth characteristics, and hospital feeding practices
**Piper 1996** [54]	USA	Cohort study	January 1989–June 1991	2372	Data from the 1988 National Maternal-Infant Health Survey	All workers	24,405	Exclusive and non-exclusive	<6 weeks; 6 weeks–3 months; after 3 months and up to 6 months	Maternal sociodemographics and employment patterns
**Scott 2006** [55]	Australia	Cohort study	September 2002–July 2003	587	Mothers contacted within the 3 days after birth from 2 maternity hospitals in Perth	All workers	23,437	Exclusive and non-exclusive	<6 months; 6–12 month	Maternal sociodemographics, employment patterns, birth characteristics, hospital feeding practices, and psychosocial factors
**Xiang 2016** [56]	Australia	Cross-sectional study	November 2010–February 2011	2300	Data from the BaselineMothers Survey	Paid employment	51,937	Exclusive and non-exclusive	<3 months; 3–6 months; <8 weeks; 9–16 weeks	Maternal sociodemographics and employment patterns
**Yimyam 1999** [57]	Thailand	Cross-sectional study	July–August 1994 and April–November 1995	295	Women approached in the growth monitoring clinic at Chiang Mai University Hospital or at Chiang Mai University’s Child Care Centre	Formal sector (public and private employee) and informal sector (pieceworker at home and self/family employed)	2845	Exclusive and non-exclusive	6 months	Maternal sociodemographics and employment patterns

## Data Availability

All relevant data were included in the paper.

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
