# Peer review of "Breastfeeding after Returning to Work: A Systematic Review and Meta-Analysis"

_ijerph, 2021, doi:10.3390/ijerph18168631_

Round 1

Reviewer 1 Report

Intro:

  • Sentence one, repetition of the word “benefits” recommend rephrase
  • Line four: change to “ with the addition of solid food”
  • Not sure what the word conciliate is meant to mean – do they mean reconcile?
  • Recommend addition of possible reasons for decreased breastfeeding rates during RTW

Methods:

  • Was the study registered or funded
  • Inclusion criteria listed but exclusion criteria not explicitly listed
  • What methods were used to assess risk of bias of included studies, I see the diagram but this should be described in the methods

Results:

  • Would remove bolding from the paragraphs under Meta-analysis on prevalence of breastfeeding at return to work
  • During the meta analysis was this separated based on months after delivery and RTW – would expect a large difference between 0-3 mo; 3-6 and after 6 months due to different factors such as establishing supply and introduction of solid foods, etc
  • Did you stratify by race, as this influences baseline breastfeeding rates

Discussion:

  • Would recommend under “other factors” discussing the existence of legislation in certain countries mandating space for breastfeeding (e.g. in the US)

Conclusion:

  • First sentence of the conclusion too strong given heterogeneity and confounding factors.

PRISMA diagram for inclusion exclusion criteria application seems to be missing from the supplementary files

Author Response

Reviewer 1

English language and style

(x) Moderate English changes required

[REPLY] Thank you for this comment. The article has been proof-read by a native English speaker to improve its readability and some misspelling.

Does the introduction provide sufficient background and include all relevant references?

(x) Can be improved

Is the research design appropriate?

(x) Can be improved

Are the methods adequately described?

(x) Can be improved

Are the results clearly presented?

(x) Can be improved

Are the conclusions supported by the results?

(x) Must be improved

[REPLY] Thank you for this comment. We tried to improve the whole article, particularly the conclusion. The conclusion now reads: “Despite the scarcity of data, the prevalence of breastfeeding at return to work is 25% and widely heterogeneous in the world. Even if economic status may play a role in breastfeeding at return to work, cultural aspects seem an important determinant, influencing public health policies and workplace breastfeeding support. We also showed the lack of data regarding breastfeeding at return to work in most countries, with no data available from some continents such as Africa, demonstrating the strong need for data in those countries to inform effective preventive strategies.”

Comments and Suggestions for Authors

Intro:

Sentence one, repetition of the word “benefits” recommend rephrase

[REPLY] Thank you for this comment. The first sentence now reads: “Breastfeeding provides multiple health advantages for the child (infections, mal-occlusion, intelligence) and their mother (breast cancer) [1–4], with economic and social benefits as well (cost savings for parents, employers, and society; parent-child relation) [3,5–7].”

Line four: change to “ with the addition of solid food”

[REPLY] Thank you for this comment. Following recommendations from Reviewer 2, we now cite exactly the statement from the WHO. The sentence now reads: “Hence, the World Health Organization (WHO) recommends “exclusive breastfeeding for the first 6 months of life and introduction of nutritionally-adequate and safe complementary (solid) foods at 6 months together with continued breastfeeding up to 2 years of age or beyond” [8].

Not sure what the word conciliate is meant to mean – do they mean reconcile?

[REPLY] Thank you for this comment. The end of the sentence now reads: “[…] for mothers who have to manage work and breastfeeding.”

Recommend addition of possible reasons for decreased breastfeeding rates during RTW

[REPLY] Thank you for this comment. The introduction now reads: “During this breastfeeding transition time, return to work (RTW) is common for mothers who have to manage work and breastfeeding. RTW represents one of the main rea-son for stopping breastfeeding [9–12]. Combining breastfeeding and work may be hard for mothers depending on their working conditions [13], socio-cultural heritage and gender role ideology [14], public health policies [15], economy and lobbies [16]. For example, in a Taiwanese study, 67% of working mothers initiated breastfeeding but only 10% continued after RTW [17]. Both the culture of work and breastfeeding differ be-tween countries – for example breastfeeding initiation may vary from 47% (Ireland) to 99% (Norway) [18] within developed European countries. In addition to breastfeeding initiation, type of breastfeeding (exclusive and non-exclusive) may also be at the inter-play between work environment and socio-cultural-economic aspects [19].”

Methods:

Was the study registered or funded

[REPLY] Thank you for this comment. The study was not registered as there were delay in procedures during the COVID-19 pandemic. The funding is mentioned before references: “Funding: This study was funded by the University Hospital of Clermont-Ferrand, France.”

Inclusion criteria listed but exclusion criteria not explicitly listed

[REPLY] Thank you for this comment. The section now reads: “The search was not limited to specific years. To be included, articles needed to describe our primary outcome variable, which was the prevalence of breastfeeding after RTW i.e. women had to RTW and studies had to report the timing of RTW. Specifically, we excluded studies when mothers did not work, or not describing breastfeeding and its timing related to RTW. Studies that were not written in English or French were also excluded, as well as qualitative studies.”

What methods were used to assess risk of bias of included studies, I see the diagram but this should be described in the methods

[REPLY] Thank you for this comment. The title for the Appendix 4 now reads: “Appendix 4. Risk of bias of included articles using the Newcastle-Ottawa Quality Assessment Scale.” The description of risk of bias of included studies in written in the section “Quality of assessment” of the Methods, as follow: “The assessment of methodological quality was performed using the Newcastle-Ottawa Scale (NOS) for cohort studies [32] and modified NOS for cross-sectional stud-ies [33]. Items assessed were selection bias (four items), comparability bias (one item), and outcome bias (3 items for cohort and 2 for cross-sectional studies). Each item was assigned a judgment of “Yes” (1 point), “No” (0 point), or “Can’t say” (0 point). Thus, the maximum score was 8 points for cohort studies and 7 points for cross sectional studies (Appendix 3 and 4). Disagreements between reviewers (G. Méchin and M. Clinchamps) were addressed by obtaining a consensus with a third author (F. Dutheil).” Please do not hesitate if you would like that we add something else.

Results:

Would remove bolding from the paragraphs under Meta-analysis on prevalence of breastfeeding at return to work

[REPLY] Thank you for this comment. Amended.

During the meta analysis was this separated based on months after delivery and RTW – would expect a large difference between 0-3 mo; 3-6 and after 6 months due to different factors such as establishing supply and introduction of solid foods, etc

[REPLY] Thank you for this comment. We totally agree with your comment that stratification by time would have strengthened the article. However, it was unfortunately not possible because of a huge heterogeneity between studies in time of return to work, without common timing (see Table 1, before last column “Timing of return to work”. The limitations section reads: “Furthermore, dates of RTW were too heterogeneous to stratify for; the lack of included studies also precluded stratification by time – both may have impacted comparisons between continents and GDP.”

Did you stratify by race, as this influences baseline breastfeeding rates

[REPLY] Thank you for this comment. We added the following sentence within the limitations section: “Stratification by ethnicity were not feasible because of lack of data, however stratification by country of breastfeeding / continent gave international comparisons and should have taken into account the influences of baseline breastfeeding rates.” Most countries included have a low immigration rate. For example, population of Asian countries should be Asiatic, so our stratification by continent should give an insight on ethnicity.

Discussion:

Would recommend under “other factors” discussing the existence of legislation in certain countries mandating space for breastfeeding (e.g. in the US)

[REPLY] Thank you for this comment. We added the following sentence in the aforementioned section: “Even if some laws promoted breastfeeding at work, such as the Federal Break Time for Nursing Mothers law requiring employers covered by the Fair Labor Standards Act (FLSA) to provide basic accommodations for breastfeeding mothers at work in the US, those laws are still not fully applicated [16] and need to be expanded worldwide.” The second section of the discussion also reads: “Breastfeeding can also be at the interplay between public health policies, economy and lobbies. In USA, the Infant Formula Council historically lobbied against the public health promotion of breastfeeding [16], even discouraging a pro-breastfeeding cam-paign in 2007 [75]. In 2009, only 23/50 states in USA encouraged workplace breastfeed-ing by adopting laws, and no state required employers to provide breastfeeding pump-ing equipment to their employees [76]. In 2011, USA ranked last out of 36 countries for its breastfeeding policy [16]. Eager to improve worksite lactation, USA launched ambitious programs [77], with reasonable break time and adequate space for nursing mothers to express milk [78].”

Conclusion:

First sentence of the conclusion too strong given heterogeneity and confounding factors.

[REPLY] Thank you for this comment. The conclusion now reads: “Despite the scarcity of data, the prevalence of breastfeeding at return to work is 25% and widely heterogeneous in the world. Even if economic status may play a role in breastfeeding at return to work, cultural aspects seem an important determinant, influencing public health policies and workplace breastfeeding support. We also showed the lack of data regarding breastfeeding at return to work in most countries, with no data available from some continents such as Africa, demonstrating the strong need for data in those countries to inform effective preventive strategies.”

PRISMA diagram for inclusion exclusion criteria application seems to be missing from the supplementary files

[REPLY] Thank you for this comment. We added “studies excluded” within the Prisma diagram (Figure 1). Sorry if we have misunderstood. Prisma diagram is Figure 1 (we are not sure about what is missing from the supplementary files i.e. Appendix).

Reviewer 2 Report

Dear authors, congratulations for your article! I think that your work contributes to the advancement of existing knowledge. Also, the quality of presentation is high and the research is well-designed and technically sound. The only thing that I would like you to check is the reference number 8, at page 1, because it seems that is not right for this sentence. Moreover, the real statement from WHO is "Exclusive breastfeeding for the first 6 months of life and introduction of nutritionally-adequate and safe complementary (solid) foods at 6 months together with continued breastfeeding up to 2 years of age or beyond." Best regards

Author Response

Reviewer 2

English language and style

(x) I don't feel qualified to judge about the English language and style

[REPLY] Thank you for this comment.

Does the introduction provide sufficient background and include all relevant references?

(x) Yes

Is the research design appropriate?

(x) Yes

Are the methods adequately described?

(x) Yes

Are the results clearly presented?

(x) Yes

Are the conclusions supported by the results?

(x) Yes

[REPLY] Thank you very much for your positive opinion.

Comments and Suggestions for Authors

Dear authors, congratulations for your article! I think that your work contributes to the advancement of existing knowledge. Also, the quality of presentation is high and the research is well-designed and technically sound. The only thing that I would like you to check is the reference number 8, at page 1, because it seems that is not right for this sentence. Moreover, the real statement from WHO is "Exclusive breastfeeding for the first 6 months of life and introduction of nutritionally-adequate and safe complementary (solid) foods at 6 months together with continued breastfeeding up to 2 years of age or beyond." Best regards

[REPLY] Thank you for very positive comment. Much appreciated. We now cite exactly the statement from the WHO. The sentence now reads: “Hence, the World Health Organization (WHO) recommends “exclusive breastfeeding for the first 6 months of life and introduction of nutritionally-adequate and safe complementary (solid) foods at 6 months together with continued breastfeeding up to 2 years of age or beyond” [8]. Reference 8 has been updated and is now: “World Health Report 2005: Make every mother and child count. Geneva: WHO, 2005”.

Reviewer 3 Report

Méchin and colleagues have admirably reviewed breastfeeding after return to work. The work on itself is novel as per my understanding, as previous reviews were only focused on interventions and policies. However, there is a lot of scope for improvement and is advised to thoroughly revise the manuscript prior to consideration in reputable IJERPH. I have suggested few importation points below despite a gross revision on self is advised.

Abstract:

Advised to be revised as per comments below in each section.

Introduction:

The introduction is very brief and lacks basic background information for the readers.

My advice would be to include the following detail: the importance of breastfeeding overall, the impact of work-related responsibilities on breastfeeding (need to conduct an extensive literature review), since the authors are stratifying it based on the continent so some literature as a background is advised, factors related to work which has a major impact on BF, need to add detail on BF initiation, type of BF (exclusive, full, and in relation to work environment), risks associated, income and social standing as a factor, etc.

The research aims need to be mentioned clearly as per the review conducted.

The rationale for systematic review, what novel information does this research generate for the global research community?

Method:

Method and result are mixed up (a lot of information of result should be in method and vice versa). Advised to restructure it accordingly. Why are the author having an inclusion/exclusion criterion in the result? It suggests that the inclusion and exclusion criteria were developed after the search was finalized. Study selection is a pivotal aspect of a systematic review and needs to be described elaboratively and concisely.

Why is table 1 in methods?

Advised to have a clear and concise study design especially inclusion and exclusion criteria for the review in the methods section.

In relation to the database, MEDLINE (OVID), Scopus, Web of Science (ISI), PsychInfo, ProQuest Central was it considered?

Were the biases based on confounding, exposure measurement, selection of participants, measurement of outcomes, missing data, and selection of reported results considered? Some of the supplementary/ Appendix covers that however it should be briefly explained in the main manuscript too.

Since the author mentions, semi-structured interviews in the study type, are the authors considering qualitative studies? If yes, advised revising the review accordingly? How was the quality of the article considered? How was the finding comparable with the other quantitative studies included?

Results:

As per earlier advice, the result itself has a plethora of information which I believe is not required in this section in results rather fits well in methods, advised to be revised accordingly.

A PRISMA diagram would be more explanative to the author.

Appendix 5 has some informative graphs that can be considered in the main manuscript in the metanalysis section.

Advised to have a section of study characteristics where table 1 fits in.

Any results on the certainty of findings?

Are there any common findings in all studies?

Metanalysis:

Please have a forest plot for the analysis of pooled prevalence.

Discussion

The discussion reads like a repetition of the results rather than a discussion of results. Advised to be grossly revised.

Appendix

As advised, appendix 5 has some informative graphs that can be considered in the main manuscript.

Please make a list of studies excluded despite meeting your criteria, if any?

Author Response

Reviewer 3

English language and style

(x) English language and style are fine/minor spell check required

[REPLY] Thank you for this comment. The article has been proof-read by a native English speaker to improve its readability and some misspelling.

Does the introduction provide sufficient background and include all relevant references?

(x) Must be improved

Is the research design appropriate?

(x) Must be improved

Are the methods adequately described?

(x) Must be improved

Are the results clearly presented?

(x) Must be improved

Are the conclusions supported by the results?

(x) Yes

[REPLY] Thank you for this comment. We tried to improve the whole article.

Comments and Suggestions for Authors

Méchin and colleagues have admirably reviewed breastfeeding after return to work. The work on itself is novel as per my understanding, as previous reviews were only focused on interventions and policies.

[REPLY] Thank you for this very positive opinion.

However, there is a lot of scope for improvement and is advised to thoroughly revise the manuscript prior to consideration in reputable IJERPH. I have suggested few importation points below despite a gross revision on self is advised.

[REPLY] Thank you for this comment. We have answered all your comments and tried to improve the whole article.

Abstract:

Advised to be revised as per comments below in each section.

[REPLY] Thank you for this comment. The research aims are now more explicit. If you have further concerns, please do not hesitate. Please note that we are limited to 250 words and that we do have 250 words in our abstract by now. Thus any additional words should be compensated by the removal of the same number of words. 

Introduction:

The introduction is very brief and lacks basic background information for the readers. My advice would be to include the following detail: the importance of breastfeeding overall, the impact of work-related responsibilities on breastfeeding (need to conduct an extensive literature review), since the authors are stratifying it based on the continent so some literature as a background is advised, factors related to work which has a major impact on BF, need to add detail on BF initiation, type of BF (exclusive, full, and in relation to work environment), risks associated, income and social standing as a factor, etc.

[REPLY] Thank you for this comment. We added some references and sentences on the importance of breastfeeding and on some aspects related to our article i.e. breastfeeding at return to work. For example, we now introduce more factors related to work such as BF initiation depending on countries, type of BF (exclusive, full, and in relation to work environment), and income and social standing as a factor. The introduction now reads: “Breastfeeding provides multiple health advantages for the child (infections, mal-occlusion, intelligence) and their mother (breast cancer) [1–4], with economic and social benefits as well (cost savings for parents, employers, and society; parent-child relation) [3,5–7]. Hence, the World Health Organization (WHO) recommends “exclusive breastfeeding for the first 6 months of life and introduction of nutritionally-adequate and safe complementary (solid) foods at 6 months together with continued breastfeeding up to 2 years of age or beyond” [8]. During this breastfeeding transition time, re-turn to work (RTW) is common for mothers who have to manage work and breastfeeding. RTW represents one of the main reason for stopping breastfeeding [9–12]. Combining breastfeeding and work may be hard for mothers depending on their working conditions [13], socio-cultural heritage and gender role ideology [14], public health policies [15], economy and lobbies [16]. For example, in a Taiwanese study, 67% of working mothers initiated breastfeeding but only 10% continued after RTW [17]. Both the culture of work and breastfeeding differ between countries – for example breastfeeding initiation may vary from 47% (Ireland) to 99% (Norway) [18] within developed European countries. In addition to breastfeeding initiation, type of breastfeeding (exclusive and non-exclusive) may also be at the interplay between work environment and socio-cultural-economic aspects [19]. However, no studies summarize differences in breastfeeding after RTW and compare countries. Conversely, women from low-income countries have difficulty combining work and breastfeeding [20], and therefore might be at risk of ceasing breastfeeding when returning to work.”

The research aims need to be mentioned clearly as per the review conducted.

[REPLY] Thank you for this comment. The research aims now reads: “Therefore, we aimed conducted a systematic review and meta-analysis to evaluate the prevalence of breastfeeding after RTW (primary aim). Secondary objectives were to evaluate differences between continents or their level of development, as well as putative influencing variables such as sociodemographic [21–23], breastfeeding support at work [24–27] or working policy [28–30]. Additionally, we evaluated the influence of the previous factors on the type of breastfeeding (exclusive or not). Please do not hesitate to give us the sentence you would like if not satisfied of our proposal.

The rationale for systematic review, what novel information does this research generate for the global research community?

[REPLY] Thank you for this comment. We added the following sentence before the aims: “Considering the importance of breastfeeding, an evidence-based study is needed to summarize the existing literature for building efficient promotion and support for breastfeeding at the workplace. Therefore, we aimed conducted a systematic review and meta-analysis to evaluate the prevalence of breastfeeding after RTW (primary aim). Secondary objectives were to evaluate differences between continents or their level of development, as well as putative influencing variables such as sociodemographic [21–23], breastfeeding support at work [24–27] or working policy [28–30]. Additionally, we evaluated the influence of the previous factors on the type of breastfeeding (exclusive or not).”

Method:

Method and result are mixed up (a lot of information of result should be in method and vice versa). Advised to restructure it accordingly. Why are the author having an inclusion/exclusion criterion in the result? It suggests that the inclusion and exclusion criteria were developed after the search was finalized. Study selection is a pivotal aspect of a systematic review and needs to be described elaboratively and concisely.

[REPLY] Thank you for this comment. Sorry for the misunderstanding. In the results section, the section “inclusion and exclusion criteria” was “Inclusion and exclusion criteria within included articles” (and not the inclusion and exclusion criteria that we used – it is the inclusion and exclusion criteria that included articles used in their studies). The title of this section has been updated accordingly to avoid any confusion for readers. This section is a usual section in systematic review. Please see our previous meta-analyses that have also this section (more than 30 articles: https://pubmed.ncbi.nlm.nih.gov/?term=dutheil+f+meta-analysis). Inclusion and exclusion criteria of our systematic review and meta-analysis has also been updated to be more precise in the Methods section. The section now reads: “The search was not limited to specific years. To be included, articles needed to describe our primary outcome variable, which was the prevalence of breastfeeding after RTW i.e. women had to RTW and studies had to report the timing of RTW. Specifically, we excluded studies when mothers did not work, or not describing breastfeeding and its timing related to RTW. Studies that were not written in English or French were also excluded, as well as qualitative studies.”

Why is table 1 in methods?

[REPLY] Thank you for this comment. Table 1 has been placed in the Methods section by IJERPH but we agree that Table 1 can also be appropriately placed in the Results section. Following suggestion of two reviewers, we propose to move Table 1 in the Results section.

Advised to have a clear and concise study design especially inclusion and exclusion criteria for the review in the methods section.

[REPLY] Thank you for this comment. The section now reads: “The search was not limited to specific years. To be included, articles needed to describe our primary outcome variable, which was the prevalence of breastfeeding after RTW i.e. women had to RTW and studies had to report the timing of RTW. Specifically, we excluded studies when mothers did not work, or not describing breastfeeding and its timing related to RTW. Studies that were not written in English or French were also excluded, as well as qualitative studies.”

In relation to the database, MEDLINE (OVID), Scopus, Web of Science (ISI), PsychInfo, ProQuest Central was it considered?

[REPLY] Thank you for this comment. Most meta-analysis use three databases, sometimes four. We considered PubMed, Cochrane Library, Embase and Base.

Were the biases based on confounding, exposure measurement, selection of participants, measurement of outcomes, missing data, and selection of reported results considered? Some of the supplementary/ Appendix covers that however it should be briefly explained in the main manuscript too.

[REPLY] Thank you for this comment. The Results section now reads: “Quality assessment of the 14 included studies, as outlined by the NOS, varied from 57.1 [57] to 100% [44], with a mean score of 81.8±7.9%. Most frequent bias were assessment of outcomes (self-reported) for cohort studies, and bias of selection, especially considering the limited sample size in some studies. There is also a lack of follow-up in cohort studies. Detailed characteristics of methodological quality assessment of each included study are available in Appendix 3 and 4. All studies mentioned ethical approval.” We also updated the Limitations section: “All meta–analyses have limitations [93]. Meta–analyses inherit the limitations of the individual studies of which they are composed and are subjected to a bias of selection of included studies. However, the use of broader keywords in the search strategy limited the number of missing studies. Despite our rigorous criteria for including studies in our meta–analysis, their quality varied. Most cross-sectional studies included in our meta-analyses described a bias of self-report such as skipping questions and in-complete information, nondisclosure, and uncertainty regarding timing of questionnaire. Though there were similarities between the inclusion criteria, they were not identical. In particular, some studies included only mothers who worked the year be-fore delivery, whereas other studies did not specify [47,49,50,54,55]. Two studies only included women who initiated breastfeeding [47,54] that may lead to a comparison bi-as however sensitivity analyses without those two studies did not influence our results. Moreover, our meta-analysis was based on a moderate number of studies, especially for exclusive breastfeeding. A strong result of our study is also the lack of breast-feeding data at RTW – some continents have even no data. Stratification by ethnicity were not feasible because of lack of data, however stratification by country of breast-feeding / continent gave international comparisons and should have taken into account the influences of baseline breastfeeding rates. Furthermore, dates of RTW were too heterogeneous to stratify for; the lack of included studies also precluded stratification by time – both may have impacted comparisons between continents and GDP.”

Since the author mentions, semi-structured interviews in the study type, are the authors considering qualitative studies? If yes, advised revising the review accordingly? How was the quality of the article considered? How was the finding comparable with the other quantitative studies included?

[REPLY] Thank you for this comment. Qualitative studies were an exclusion criterion (explicitly mentioned in Figure 1). To avoid any confusion, we added the following sentence in the Methods section: “Studies that were not written in English or French were also excluded, as well as qualitative studies.”

Results:

As per earlier advice, the result itself has a plethora of information which I believe is not required in this section in results rather fits well in methods, advised to be revised accordingly.

[REPLY] Thank you for this comment. Sorry for the misunderstanding. In the results section, the section “inclusion and exclusion criteria” was “Inclusion and exclusion criteria within included articles” (and not the inclusion and exclusion criteria that we used – it is the inclusion and exclusion criteria that included articles used in their studies). The title of this section has been updated accordingly to avoid any confusion for readers. The sections of the Results are usual sections in systematic reviews. Please see our previous meta-analyses that have also those sections (more than 30 articles: https://pubmed.ncbi.nlm.nih.gov/?term=dutheil+f+meta-analysis). If you would like, parts of results can be placed as supplementary materials (Appendix). Please see for example the following articles that have those sections in supplementary materials: Navel V, Malecaze J, Pereira B, Baker JS, Malecaze F, Sapin V, Chiambaretta F, Dutheil F. Oxidative and antioxidative stress markers in keratoconus: a systematic review and meta-analysis. Acta Ophthalmol. 2020 Dec 23. doi: 10.1111/aos.14714; Dutheil F, Aubert C, Pereira B, Dambrun M, Moustafa F, Mermillod M, Baker JS, Trousselard M, Lesage FX, Navel V. Suicide among physicians and health-care workers: A systematic review and meta-analysis. PLoS One. 2019 Dec 12;14(12):e0226361. doi: 10.1371/journal.pone.0226361; Dutheil F, Baker JS, Mermillod M, De Cesare M, Vidal A, Moustafa F, Pereira B, Navel V. Shift work, and particularly permanent night shifts, promote dyslipidaemia: A systematic review and meta-analysis. Atherosclerosis. 2020 Nov;313:156-169. doi: 10.1016/j.atherosclerosis.2020.08.015.

We can do it here too if you would like (moving part of results in Appendix).

A PRISMA diagram would be more explanative to the author.

[REPLY] Thank you for this comment. PRISMA diagram has been updated as Figure 1. Please feel free to give comments if you would like more changes.

Appendix 5 has some informative graphs that can be considered in the main manuscript in the metanalysis section.

[REPLY] Thank you for this comment. Appendix 5 is now Figure 4.

Advised to have a section of study characteristics where table 1 fits in.

[REPLY] Thank you for this comment. Table 1 has been placed in the Methods section by IJERPH but we agree that Table 1 can also be appropriately placed in the Results section. Following suggestion of two reviewers, we propose to move Table 1 in the Results section.

Any results on the certainty of findings?

[REPLY] Thank you for this comment. The Results section now reads: “Quality assessment of the 14 included studies, as outlined by the NOS, varied from 57.1 [57] to 100% [44], with a mean score of 81.8±7.9%. Most frequent bias were assessment of outcomes (self-reported) for cohort studies, and bias of selection, especially considering the limited sample size in some studies. There is also a lack of follow-up in cohort studies. Detailed characteristics of methodological quality assessment of each included study are available in Appendix 3 and 4. All studies mentioned ethical approval.” We also updated the Limitations section: “All meta–analyses have limitations [93]. Meta–analyses inherit the limitations of the individual studies of which they are composed and are subjected to a bias of selection of included studies. However, the use of broader keywords in the search strategy limited the number of missing studies. Despite our rigorous criteria for including studies in our meta–analysis, their quality varied. Most cross-sectional studies included in our meta-analyses described a bias of self-report such as skipping questions and in-complete information, nondisclosure, and uncertainty regarding timing of questionnaire. Though there were similarities between the inclusion criteria, they were not identical. In particular, some studies included only mothers who worked the year be-fore delivery, whereas other studies did not specify [47,49,50,54,55]. Two studies only included women who initiated breastfeeding [47,54] that may lead to a comparison bi-as however sensitivity analyses without those two studies did not influence our results. Moreover, our meta-analysis was based on a moderate number of studies, especially for exclusive breastfeeding. A strong result of our study is also the lack of breast-feeding data at RTW – some continents have even no data. Stratification by ethnicity were not feasible because of lack of data, however stratification by country of breast-feeding / continent gave international comparisons and should have taken into account the influences of baseline breastfeeding rates. Furthermore, dates of RTW were too heterogeneous to stratify for; the lack of included studies also precluded stratification by time – both may have impacted comparisons between continents and GDP.”

Are there any common findings in all studies?

[REPLY] Thank you for this comment. As we specified in the first sentence of the meta-analysis section, there is “an important heterogeneity (I2=98.6%) – prevalence of breastfeeding after RTW ranging from 2% [43] to 61% [40].” Then we detailed the factors that we were able to statistically analyze and that explained this heterogeneity (continents, GDP, etc.).

Metanalysis:

Please have a forest plot for the analysis of pooled prevalence.

[REPLY] Thank you for this comment. We added the forest plot for the meta-analysis of prevalence of breastfeeding at return to work, stratified by continents (Appendix 5) and stratified by GDP per capita (Appendix 6).

Discussion

The discussion reads like a repetition of the results rather than a discussion of results. Advised to be grossly revised.

[REPLY] Thank you for this comment. We did not have included our results in the discussion. All numbers are from other studies to put our results in perspective of literature. We discuss our main outcomes i.e. prevalence of breastfeeding, the influence of continents and cultural aspect of breastfeeding, as well as the influence of economy and public health policies. We agree that our discussion might have been too short. Therefore, we added in the discussion several other ideas, such as the role of family support, gender role ideology, social network, the role of workplace factors, flexible working arrangements, and national family policies. The discussion now includes the following sentences: “[…] It is known that the cultural aspect is very important for breastfeeding uptake [19]. Mothers' mothers have a strong positive attitude towards breastfeeding when they are positively reinforced or supported [66]. Notably, highly educated Chinese grandmothers were associated with decreased exclusive breastfeeding in their daughters [67]. This fact could be linked with gender role ideology that varies markedly across countries [68]. Moreover, social and cultural attitudes have an impact on the representation of breastfeeding within and between different countries/continents. A meta-analysis found that community-based interventions, including group counselling or education and social mobilization, with or without mass media, were effective at increasing timely breastfeeding initiation by 86% and exclusive breastfeeding by 20% [19]. […] Maternity leave also positively impacts breastfeeding duration [10,12,73,74]. A recent review shows a positive relationship between maternity leave length and breastfeeding duration [75]. […] More broadly, the Lancet breastfeeding series highlighted that the promotion of breastfeeding is a collective societal responsibility and not the sole responsibility of an individual woman [19]. One of the six call to action points was to foster positive societal attitudes towards breastfeeding like adequate maternity leave and the opportunity to breastfeed or express milk in the workplace [19]. […] Flexibility in working schedules may be associated with breastfeeding [87]. Despite no studies, the acceptance of teleworking following the COVID-19 pandemic could also help women to breastfeed [88]. Interestingly, the guarantee of paid breastfeeding breaks for at least 6 months has been shown to be associated with an increase of nearly 9% of exclusive breastfeeding [89]. […] Even if some laws promoted breastfeeding at work, such as the Federal Break Time for Nursing Mothers law requiring employers covered by the Fair Labor Standards Act (FLSA) to provide basic accommodations for breastfeeding mothers at work in the USA, those laws are still not fully applicated [16] and need to be expanded worldwide. There is very limited or inexistent literature on the putative sociodemographic and clinical factors linked with breastfeeding at RTW. […]” We also added more than ten references, mainly systematic reviews, meta-analysis, or studies from high quality journals such as The Lancet.

Appendix

As advised, appendix 5 has some informative graphs that can be considered in the main manuscript.

[REPLY] Thank you for this comment. Appendix 5 is now Figure 4.

Please make a list of studies excluded despite meeting your criteria, if any?

[REPLY] Thank you for this comment. There are no studies that met our inclusion criteria but that we excluded (all studies meeting our inclusion criteria were included).

Reviewer 4 Report

Thank you for giving me the opportunity to review the paper. Overall, this paper is important and fills a crucial research gap. There are some minor comments.

(1) In the introduction, you mentioned the role of culture in shaping breastfeeding. You need to be more specific, what kind of culture, and how and why it can affect breastfeeding.

(2) In the conclusion, you summarized the potential impacts of different factors on breastfeeding. You only roughly distinguished between cultural and other factors. I think there are more factors that worth particular attention. For example, whether women receive family support can influence their breastfeeding status.

(3) Following the previous comment, I think gender role ideology is a salient factor that can determine whether women continue breastfeed their children. Women with traditional gender role ideology may be more likely to stay at home after giving birth to a child.  I think this factor is worth particular attention. Some possible works can be consulted.

Knight, C.R. and Brinton, M.C., 2017. One egalitarianism or several? Two decades of gender-role attitude change in Europe. American Journal of Sociology122(5), pp.1485-1532.

Wang, S., 2019. The role of gender role attitudes and immigrant generation in ethnic minority women’s labor force participation in Britain. Sex Roles80(3), pp.234-245.

(4) Also, peer influence, social network and neighbourhood may also be important in affecting women's decision of breastfeeding.

(5) You briefly mentioned the role of work place factors, but I think flexible working arrangements or national family policies are of great importance for women's ability to breastfeed their children and worth more attention. This is especially the case during the pandemic.

Author Response

Reviewer 4

English language and style

(x) I don't feel qualified to judge about the English language and style

[REPLY] Thank you for this comment.

Does the introduction provide sufficient background and include all relevant references?

(x) Can be improved

Is the research design appropriate?

(x) Yes

Are the methods adequately described?

(x) Yes

Are the results clearly presented?

(x) Yes

Are the conclusions supported by the results?

(x) Can be improved

[REPLY] Thank you for this comment. We particularly improved the introduction and tempered our conclusion.

Comments and Suggestions for Authors

Thank you for giving me the opportunity to review the paper. Overall, this paper is important and fills a crucial research gap. There are some minor comments.

[REPLY] Thank you for your very positive comment.

(1) In the introduction, you mentioned the role of culture in shaping breastfeeding. You need to be more specific, what kind of culture, and how and why it can affect breastfeeding.

[REPLY] Thank you for this comment. We added some sentences in the introduction. The introduction now reads: “During this breastfeeding transition time, return to work (RTW) is common for mothers who have to manage work and breastfeeding. RTW represents one of the main rea-son for stopping breastfeeding [9–12]. Combining breastfeeding and work may be hard for mothers depending on their working conditions [13], socio-cultural heritage and gender role ideology [14], public health policies [15], economy and lobbies [16]. For ex-ample, in a Taiwanese study, 67% of working mothers initiated breastfeeding but only 10% continued after RTW [17]. Both the culture of work and breastfeeding differ be-tween countries – for example breastfeeding initiation may vary from 47% (Ireland) to 99% (Norway) [18] within developed European countries. In addition to breastfeeding initiation, type of breastfeeding (exclusive and non-exclusive) may also be at the inter-play between work environment and socio-cultural-economic aspects [19].”

(2) In the conclusion, you summarized the potential impacts of different factors on breastfeeding. You only roughly distinguished between cultural and other factors. I think there are more factors that worth particular attention. For example, whether women receive family support can influence their breastfeeding status.

(3) Following the previous comment, I think gender role ideology is a salient factor that can determine whether women continue breastfeed their children. Women with traditional gender role ideology may be more likely to stay at home after giving birth to a child.  I think this factor is worth particular attention. Some possible works can be consulted.

Knight, C.R. and Brinton, M.C., 2017. One egalitarianism or several? Two decades of gender-role attitude change in Europe. American Journal of Sociology, 122(5), pp.1485-1532.

Wang, S., 2019. The role of gender role attitudes and immigrant generation in ethnic minority women’s labor force participation in Britain. Sex Roles, 80(3), pp.234-245.

(4) Also, peer influence, social network and neighbourhood may also be important in affecting women's decision of breastfeeding.

(5) You briefly mentioned the role of work place factors, but I think flexible working arrangements or national family policies are of great importance for women's ability to breastfeed their children and worth more attention. This is especially the case during the pandemic.

[REPLY for (2) to (5)] Thank you for this relevant comment. We added in the discussion the role of family support, gender role ideology, social network, the role of workplace factors, flexible working arrangements, and national family policies. The discussion now reads: “[…] It is known that the cultural aspect is very important for breastfeeding uptake [19]. Mothers' mothers have a strong positive attitude towards breastfeeding when they are positively reinforced or supported [66]. Notably, highly educated Chinese grandmothers were associated with decreased exclusive breastfeeding in their daughters [67]. This fact could be linked with gender role ideology that varies markedly across countries [68]. Moreover, social and cultural attitudes have an impact on the representation of breastfeeding within and between different countries/continents. A meta-analysis found that community-based interventions, including group counselling or education and social mobilization, with or without mass media, were effective at increasing timely breastfeeding initiation by 86% and exclusive breastfeeding by 20% [19]. […] Maternity leave also positively impacts breastfeeding duration [10,12,73,74]. A recent review shows a positive relationship between maternity leave length and breastfeed-ing duration [75]. […] More broadly, the Lancet breastfeeding series highlighted that the promotion of breastfeeding is a collective societal responsibility and not the sole responsibility of an individual woman [19]. One of the six call to action points was to foster positive societal attitudes towards breastfeeding like adequate maternity leave and the opportunity to breastfeed or express milk in the workplace [19]. […] Flexibility in working schedules may be associated with breastfeeding [87]. Despite no studies, the acceptance of teleworking following the COVID-19 pandemic could also help women to breastfeed [88]. Interestingly, the guarantee of paid breastfeeding breaks for at least 6 months has been shown to be associated with an increase of nearly 9% of exclusive breastfeeding [89]. […] Even if some laws promoted breastfeeding at work, such as the Federal Break Time for Nursing Mothers law requiring employers covered by the Fair Labor Standards Act (FLSA) to provide basic accommodations for breastfeeding mothers at work in the USA, those laws are still not fully applicated [16] and need to be expanded worldwide. There is very limited or inexistent literature on the putative sociodemographic and clinical factors linked with breastfeeding at RTW. […]”  We did not add the idea of family support in the conclusion as it is not supported by our work. We do not have such data in the included articles used in our meta-analysis. We added all the suggested references plus several others.

Round 2

Reviewer 3 Report

Most of the comments were addressed in a structured way, I believe the manuscript looks in better form and is publishable.

However, in the version that is in the reviewer's MDPI portal, fig1-4 was not visible. As per the MDPI author's guideline figure should be placed within the main manuscript. Please adhere to the MDPI guidelines.

Forest plot as a supplementary material uplifted the quality of the review. 

Additionally in the Discussion, your finding continent-wise was interesting especially in Oceania compared to other continents. Furthermore, I would advise critical discussion on your findings based on recent literature would make your finding more relevant for numerous policymakers and related stakeholders. some examples of recent studies in Oceania are 

doi: 10.1136/bmjopen-2018-026234

https://doi.org/10.3390/ijerph17155384

Author Response

Reviewer 3

English language and style are fine/minor spell check required

[REPLY] Thank you for this comment. Several authors are native English, and the article has also been proof-read by another native English. If you still find some minor misspellings, may you indicate which. Many thanks.

Most of the comments were addressed in a structured way, I believe the manuscript looks in better form and is publishable.

[REPLY] Thank you for your positive opinion.

However, in the version that is in the reviewer's MDPI portal, fig1-4 was not visible. As per the MDPI author's guideline figure should be placed within the main manuscript. Please adhere to the MDPI guidelines.

[REPLY] Thank you for this comment. The Figures are now within the manuscript and adhere to the MDPI guidelines.

Forest plot as a supplementary material uplifted the quality of the review.

[REPLY] Thank you for your positive opinion.

Additionally in the Discussion, your finding continent-wise was interesting especially in Oceania compared to other continents. Furthermore, I would advise critical discussion on your findings based on recent literature would make your finding more relevant for numerous policymakers and related stakeholders. Some examples of recent studies in Oceania are

doi: 10.1136/bmjopen-2018-026234

https://doi.org/10.3390/ijerph17155384

[REPLY] Thank you for this relevant comment. We have added the two suggested references with one sentence in the section 2 and one sentence in the section 3 of the discussion (so that now the three sections of the discussion have 33 lines each).
